# Ordering and Dynamics of Interacting Colloidal Particles under Soft Confinement

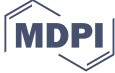

**Salvador Herrera-Velarde** [1] , **Edith C. Euán-Díaz** [2] **and Ramón Castañeda-Priego** [3,*]

1   Subdirección de Investigación y Posgrado, Tecnológico Nacional de México, Instituto Tecnológico Superior de Xalapa, Veracruz 91096, Mexico; salvador.hv@xalapa.tecnm.mx
2   EBRAINS AISBL/IVZW, Troonstraat 98, 1050 Brussels, Belgium; edith.euan-diaz@ebrains.eu
3   Department of Physical Engineering, Division of Sciences and Engineering, Campus León, University of Guanajuato, Loma del Bosque 103, León 37150, Mexico
\*   Correspondence: ramoncp@fisica.ugto.mx

**Abstract:** Confinement can induce substantial changes in the physical properties of macromolecules in suspension. Soft confinement is a particular class of restriction where the boundaries that constraint the particles in a region of the space are not well-defined. This scenario leads to a broader structural and dynamical behavior than observed in systems enclosed between rigid walls. In this contribution, we study the ordering and diffusive properties of a two-dimensional colloidal model system subjected to a one-dimensional parabolic trap. Increasing the trap strength makes it possible to go through weak to strong confinement, allowing a dimensional transition from two- to one-dimension. The non-monotonic response of the static and dynamical properties to the gradual dimensionality change affects the system phase behavior. We find that the particle dynamics are connected to the structural transitions induced by the parabolic trap. In particular, at low and intermediate confinement regimes, complex structural and dynamical scenarios arise, where the softness of the external potential induces melting and freezing, resulting in faster and slower particle diffusion, respectively. Besides, at strong confinements, colloids move basically along one direction, and the whole system behaves structurally and dynamically similar to a one-dimensional colloidal system.

**Keywords:** colloids; confinement; ordering; locomotion; Brownian dynamics





## 1. Introduction

Confining a many-body system into a small volume or area affects its physical properties [1–5]. Confinement is present in several natural and artificial systems of great scientific and technological relevance [2,6]. The degree of confinement handles many of the observed phenomena in such systems. However, other effects can be attributed solely to the intrinsic confinement characteristics: topographical (hard) or energetic (soft) [7].

A system with rigid boundaries implies a hard external potential, i.e., a potential which, according to the particle–wall separation, is either zero or infinity, thus, describing a steric volume exclusion between the particles and the confining wall [8]. On the other hand, soft confinement refers to a smooth external potential [7,8], i.e., a continuous potential that is well-defined as the particle–wall separation systematically decreases. Besides, soft confinement also entails the lack of a well-defined width of the confining region, i.e., there is not a prescribed (rigid-like) boundary that defines the available space to the particles. This kind of confinement induces different structural transitions and dynamical scenarios than those already observed with rigid confinement [6–10].

Most of the studies on confinement effects rely on scenarios with fixed confining walls, i.e., where the region that constrains the motion of the particles has a well-defined volume or area. For example, there exist several works related to hard and soft particle crystallization under rigid confinement, see, e.g., [2–7,11] and references therein. The layering and dynamics of $q2D$ superparamagnetic colloids enclosed between hard walls

have been studied experimentally and using Brownian dynamics simulations [12–15]. However, recent works have highlighted the importance of the role played by the softness of the confining mechanism [6–10,16,17]. For example, a colloidal system whose particles interact with a potential that has a short-ranged attraction and a long-ranged repulsion and confined by a harmonic-potential experiences a self-assembly process that leads to exciting and complex structural patterns [18,19].

Nowadays, there are many colloidal systems that involve or model some form of soft confinement. From an experimental point of view, a typical example consists of charged glass plates that act as soft confinement to charged colloids [7,8,20]. Flexible polymers end-grafted at the walls could act as a soft repulsive wall [17]. From a theoretical perspective, a model to describe a soft repulsion is the WCA-like potential between a wall and particles [17], or when a harmonic well confines or restricts the area or volume available to the particles [6,10,16]. These soft potentials can be systematically tuned to lead to either a weak or strong degree of confinement. Therefore, by merely playing with the softness of the potential, it is possible to suppress or enhance a certain kind of structural ordering with significant implications on the particle dynamics [16]. We refer to Reference [6] and references therein for more specific colloidal systems and nanotechnology examples where confinement plays an important role.

When one can manipulate the confining boundaries of the system, it is also possible to modify the energy landscape that the particles experience. Hence, this opens up new possibilities for self-assembling and the rational designing of mesoscopic clusters with a diversity of mechanical and optical properties [21]. Nowadays, optical traps and colloidal systems are clean and excellent model systems to explore such exciting possibilities [22].

Some recent works focused on the structural scenarios that emerge under the action of a soft potential. For example, the effect on the structure of a hard-sphere fluid near a soft wall [17]; the role played by the softness of a harmonic potential on the layering, freezing, and melting of hard spheres [6]; crystallization of soft spheres in soft confinement [7]. However, most of these studies concentrated on $q2D$ or $3D$ systems; much less explored is the situation of $2D$ or $q1D$ systems under soft confining potentials [9,10].

In an earlier contribution [23], we theoretically investigated the relationship between the structure and the dynamical behavior of a colloidal system confined by a parabolic potential whose constituents interact with a long-ranged potential, namely, a dipolar-like interaction. In such a contribution, we emphasized the structural transitions undergone by the colloidal system in a very narrow window of values of the $k$-parameter, which defines the softness of the potential or the degree of confinement. In this work, we study the transition from $2D$ to $1D$ of soft particles under soft confinement by means of computer simulations. We extend our previous study to a different interval of $k$-values, particularly to the regime of weak confinement. Essentially, the first stage of the transition from $2D$ to $q1D$ is analyzed in detail; this transition was not explored systematically in our previous work [23].

Furthermore, for the sake of the discussion, in this article, we have introduced a new set of observables that clarifies in a better way the connection between the local structure and its implication on the dynamical behavior. The bond-orientational correlation function quantifies the role of the soft boundary to inhibit or promote local hexagonal ordering [16,24]. The distribution and ordering of particles are quantified using the pair distribution function [25,26]. Finally, the so-called self-part of the intermediate scattering function describes the particle dynamics and the multiple relaxation regimes [16,26].

The parabolic potential strength is systematically tuned from low to high values (weak to strong confinement) to induce a dimensional transition. During such transition, the particle density (packing fraction) is kept constant. Hence, our study mainly focuses on the evolution of the equilibrated structure and its connection with particle dynamics. The results highlight the non-monotonic structure and dynamical behavior driven by the soft confinement during the transition from a $2D$ system close to a spontaneous crystallization to a $1D$ highly diluted colloidal system.

The manuscript is organized as follows. In Section 2, we provide some details of the model system, namely, the inter-particle interaction and the external parabolic potential. Section 3 presents a brief discussion of the standard Ermak–McCammon algorithm to carry out Brownian dynamics simulations. In Section 4, we present the main results and discussions on the structure and its relationship with the particle dynamics. Finally, a summary of the main results is included in Section 5.

## 2. Pair Potentials and External Parabolic Field

In this work, we consider colloids interacting with a repulsive screened Coulomb potential, typically known as Yukawa pair potential [27]. For further details and related parameters, see, e.g., References [25,27–29]. The average packing fraction used during the simulations is $\phi = \pi \rho a^2 = 0.1766$, where $\rho$ is the particle number density and $a$ the colloidal radius. We use the same value of $\phi$ regardless of the stiffness parameter value, $k$.

Experimentally, constraining a colloidal suspension between parallel plates leads to hard or soft confinement. Uncharged plates provide rigid confinement while regulating the superficial charge on the confining plates leads to long-ranged repulsions [7]. Besides, varying the solvent salt concentration creates a system of soft particles in soft confinement [8,20]. In the present work, to constrain the particles in one direction, we use the same external parabolic potential as in Reference [23]; $u_{ext}(y) = \frac{1}{2}ky^2$. This soft potential acts on each particle along the $y$-direction. In Reference [30], authors proposed a model for macroions confined between two parallel neutral walls. The charge distribution inside the plates generates a quadratic potential, which is analogous to the harmonic potential here considered. However, we do not attempt to model any specific theoretical or experimental system.

A soft potential does not impose a rigid boundary, however, variation of the stiffness parameter $k$ allows going through weak to strong confinement. Hence, $k^* \equiv ka^2/k_BT$ is the main physical control parameter in our analysis, with $k_B$ and $T$ being the Boltzmann constant and the absolute temperature, respectively. For further details, see References [6,31]. From now on, we will omit the asterisk in $k$. Thus, any reference to a particular $k$-value implies the dimensionless value.

## 3. Brownian Dynamics Simulation and Physical Observables

All calculations were carried out using Brownian dynamics (BD) computer simulations. We use the same protocol as described in Reference [23], but the present work does not consider hydrodynamic interactions. The dynamics of the system is then simulated as follows. At the beginning, the external potential is switched off ($k = 0$), and $N = 625$ particles are distributed in random initial positions in a square box of dimensions ($L_x = L_y = L$); periodic boundary conditions on each direction are applied. The particles move until the equilibrium configuration is reached. After this step, we increase the stiffness parameter in small steps $\Delta k = 0.1$. Then, particles move at least $5 \times 10^3$ time steps, where the minimum image convention and periodic boundary conditions are applied only in the $x$-direction. After that, to monitor the total energy per particle and the displacements along the $y$-direction, another equilibration cycle is performed. The size of this cycle extends until the total potential energy $U(\mathbf{r}^N, t)$ reaches a constant value. At the end of this cycle, we assign a new value to the box length $L_y$, related to the maximum displacement of particles in the $y$-direction. Using this new value $L_y^{new}$, the simulation box size is modified to keep the particle density constant, namely, $L_x = N/(\rho L_y^{new})$. Finally, particles are randomly placed in the new simulation box, and an additional equilibration cycle is carried out. This procedure is repeated for each increment of $k$. The largest value of $k$ used in this work is 30, which corresponds to the strong confinement regime, i.e., when the system behaves as a $1D$ system.

The previous methodology allows us the slow relaxation of the system, which prevents that particles get artificially trapped by the sudden exposition to the external potential. In most of the simulations, we use $N = 625$ particles; however, some simulations have been performed with a larger number of particles to discard size effects. The reduced time

step, $\Delta t$, is chosen as $\Delta t D_0 / a^2 = 2 \times 10^{-4}$, where $t$ is the time and $D_0 = k_B T / 6\pi \eta a$ is the Stokes–Einstein diffusion coefficient [26], with $\eta$ being the solvent shear viscosity.

### 3.1. Structural Observables

We monitor the structural evolution of the colloidal system through the pair correlation function $g(x)$ and the static structure factor $S(q_x)$ [5,26,28]. Both observables are of great interest because they determine the ordering and characteristic length scales of the dispersion. Besides, they are experimentally accessible and allow us a direct comparison with simulation results. We also compute the probability of finding a particle along the $y$-direction, $P(y)$.

The above physical observables describe the spatial distribution of particles inside the confining region. However, recent contributions have highlighted that local hexagonal ordering depends on the confining boundary softness [21]. Here, hexagonal local ordering relative to a given particle $i$ is quantified using the local bond orientational order parameter, $\psi_{6i}$, defined as $\psi_{6i} = \frac{1}{N_i} \sum_{j=1}^{N_i} e^{i6\theta_{ij}}$, where $N_i$ is the coordination number (number of the nearest neighbors) or particle $i$, $j$ labels its neighbors, and $\theta_{ij}$ is the angle between a reference axis and the bond joining particles $i$ and $j$ [16,32]. To identify particle neighbors, we employ a closest distance criterion. Thus, the global orientational order parameter is given by [32],

$$\Psi_6 = \left| 1/N \sum_{i=1}^{N} \psi_{6i} \right|, \tag{1}$$

which is the average of the local order parameters overall $N$ particles. This parameter characterizes the hexagonal symmetry in $2D$. $\Psi_6 = 1$ for a perfect triangular lattice, and $\Psi_6 = 0$ for a random liquid [13,32]. The function $g_6(r)$ is used to analyze the orientational correlation, and it is defined as [24],

$$g_6(r) = \left\langle \psi_6^*(\mathbf{r}') \psi_6(\mathbf{r}' - \mathbf{r}) \right\rangle, \tag{2}$$

where $\psi_6(\mathbf{r})$ is the local bond-orientational order parameter at the position $\mathbf{r}$. The orientational correlation function $g_6$ approaches to a constant in the solid phase, decays algebraically in the hexatic phase, and behaves exponentially in the liquid phase [24].

### 3.2. Dynamical Observables

Dynamical properties, such as the mean-square displacement (MSD), are calculated based on the equilibrium particle trajectories. Because the coupling with the parabolic potential introduces an anisotropic dynamical behavior, it is more illustrative to compute the contributions to the MSD separately. Hence, $Wx(t)$ and $Wy(t)$ denote the MSD perpendicular and parallel to the confinement, respectively [29].

Particle dynamics are also characterized through the self-part of the intermediate scattering function (sISF), defined by [16,26],

$$F_s(\mathbf{q}, t) = \frac{1}{N} \left\langle \sum_{i=1}^{N} \exp(i\mathbf{q} \cdot [(\mathbf{r}_i(t) - \mathbf{r}_i(0)]) \right\rangle, \tag{3}$$

which measures the decay in time of particle correlations at wavelength $2\pi/q$. For diluted fluids, this function approaches to an exponential decay, and the relaxation time for spatial inhomogeneities with wavenumber $q = \pi$, denoted by $\tau_\alpha$, is defined by $Fs(q, \tau_\alpha) = 1/e$. Similar to the analysis for the MSD, we focus on the decay of the sISF along the unconfined sIFS$_x$ and confined direction sIFS$_y$.

For the sake of the discussion, from now on, we describe any physical quantity in reduced units: $r^* \equiv r/a$, $t^* \equiv t D_0 / a^2$, $W^*(t) \equiv W(t)/a^2$, although we will omit the asterisk.

## 4. Results and Discussion

### 4.1. Structural Transitions: Analysis and Discussion

The response of the colloidal system to the soft confining potential is to form structural units of parallel chains of particles, i.e., lane-like formation [33–35]. As the strength of the parabolic potential increases, the system responds in a non-monotonic way. We correlate such behavior with the absence of hard-boundaries. Instead of a hard-wall, particles belonging to the top and bottom channels create effective deformable boundaries. Of course, strictly speaking, these boundary particles are not responsible for the confinement, but somehow, they allow the particle rearrangement inside the harmonic potential.

We do not associate the observed structural transitions with thermodynamic state transitions. For each $k$ value, the temporal evolution of the energy is monitored (data not shown). We have noticed that once the system reaches equilibrium after a thermalization process, the energy per particle fluctuates very little around a constant value.

#### 4.1.1. System Dimension

The transition from $2D$ to $1D$ is achieved by gradually increasing the value of $k$ following the protocol described in Section 3. For each increment of $k$, simulations are performed at constant density (or packing fraction), where a decrease in $L_y$ is accompanied by a corresponding increase in $L_x$. The box length changes along the direction of confinement are directly related to the particle maximum displacements in the $y$-direction. The most distant particles to the center of the box along the $y$-direction create a sort of flexible boundary. In this work, flexible confinement means that such particles (red particles in the snapshots of Figure 1) display more significant position fluctuations because their movements are not restricted by the presence of a hard-wall or periodic image particles; besides, they do not diffuse to the bulk. Hence, although particles belonging to the boundaries channels serve as a flexible wall, we have fixed the box dimensions in both $x$- and $y$-directions. The use of the same concentration avoids effects associated with density variations. Hence, the competition between the inter-particle potential and the external field will be the main physical mechanism behind the observed structure and the particle dynamics.

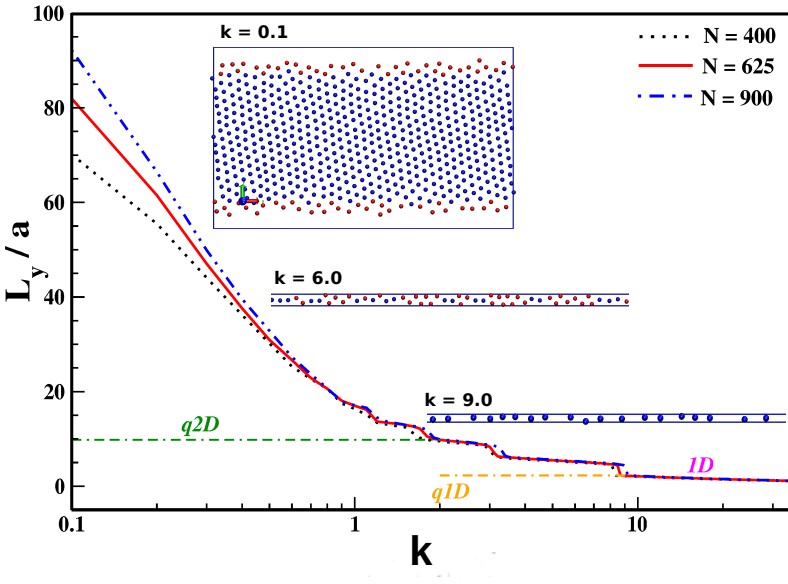

**Figure 1.** Box length in the $y$-direction as a function of the trap stiffness $k$ for three different system sizes. Snapshots were created with *VMD* [36] and represent typical configurations of particles in $q2D$, $q1D$, and $1D$ geometries. The confining harmonic potential acts on each particle individually, but boundary particles (at a given instant of time) are colored in red, and those forming the bulk in blue. Dashed lines represent approximate regions in $L_y$ and $k$ where the system exhibits a dimensionality transition.

We also monitor the dimensional transition along the direction of confinement (Figure 1). To be sure that the pathway from 2*D* to 1*D* does not include finite-size artifacts, simulations with three different box sizes were performed, i.e., different number of particles ($N = 400, 625, 900$). As shown in Figure 1, the path seems to be independent of the system size. The most dramatic changes occur in the interval of $0.1 \leq k \leq 1$, where the system transforms from 2*D* to *q*1*D*. In the region from $1.0 \leq k \leq 8.0$, a transition from *q*1*D* to 1*D* appears. For values larger than $k = 9.0$, the system has reached the limit of a 1*D* channel. In particular, for $k \geq 10$, motion along *y*-direction has been completely suppressed, and the center of mass of each particle has fluctuations in the *y*-direction smaller than the particle size.

The probability distribution of finding a particle along the *y*-direction, $P(y)$, has unique features that allow us to identify and describe a dimensional transition. Figure 2 shows the $P(y)$ for all *k*-values here studied. From there, one can deduce that particles are located, on average, in those places indicated by the maxima of $P(y)$. The presence of well-defined minima and maxima in $P(y)$ also indicates the formation of well-defined strata [2,5]. The difference in the height between a maximum and a minimum provides an estimation of the energetic barrier that a particle has to overcome to diffuse it among different strata [23].

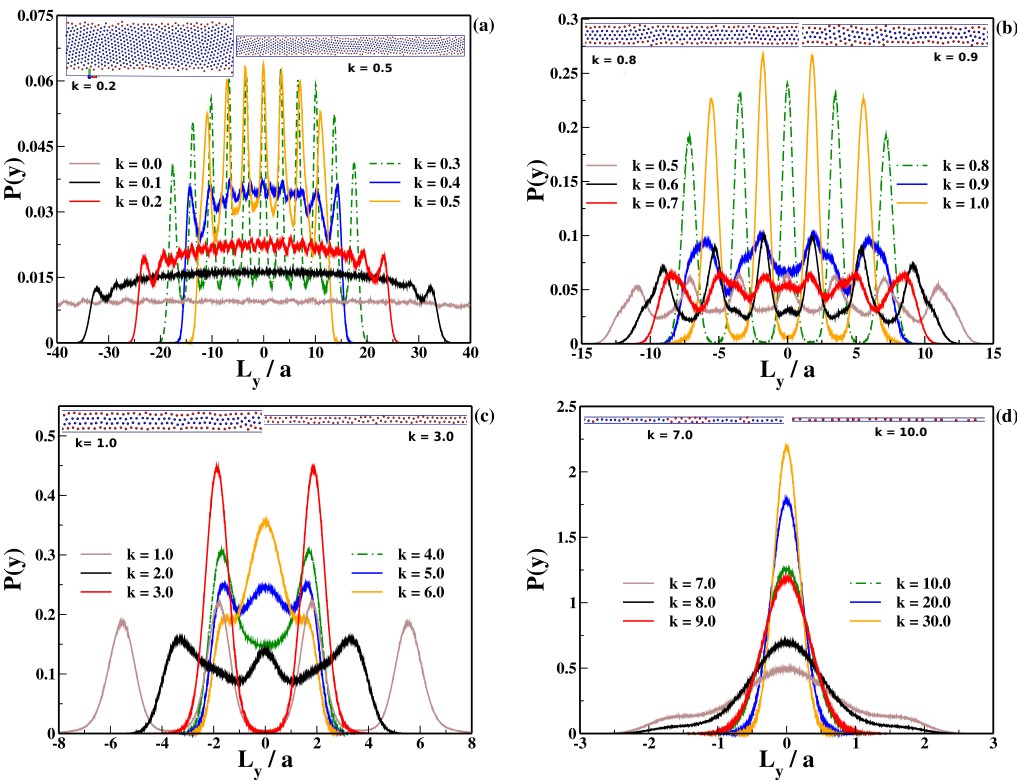

**Figure 2.** Probability distribution of finding a particle along the direction of confinement for (**a**) and (**b**) small, (**c**) intermediate and (**d**) strong couplings with the parabolic potential. The evolution of the number of peaks in the $P(y)$ gives information of the structural transition that the system undergoes from an open 2*D* system ($k = 0$) to a *q*2*D* ($0 < k \leq 1$) to *q*1*D* ($1 < k < 9$), to the final 1*D*-single-file configuration ($k \geq 9.0$).

For $k = 0$, particles distribute homogeneously; thus, the probability $P(y)$ has a constant value. For $k < 0.3$, the coupling with the external potential is very weak, however, in Figure 2a, it is possible to appreciate a change in the ordering of the colloids. Particles can move in both *x*- and *y*-directions, but some order starts to appear. In particular, for $k = 0.3$ and $0.5$, the system experiences a rearrangement of particles forming a well-defined number of channels. However, surprisingly, for $k = 0.4$, the system exhibits a kind of

re-entrant behavior, i.e., the strata are not well-defined, indicating a loss of correlation in the $y$-direction.

If one further increases the coupling with the external potential, the system exhibits once again a re-entrant structural behavior, now in the interval $0.6 \leq k \leq 1.0$ (Figure 2b). For example, when $k = 0.6, 0.8$, and $1.0$, the system exhibits well-defined strata; in contrast, when $k = 0.7$ and $0.9$, the strata dissolve in less localized configurations. Such behavior means that the system undergoes a loss of correlation along the $y$-direction. The order–disorder interconverting configurations directly impact the energetic barrier size, which becomes a non-monotonic function of the trap stiffness. Then, it is clear that even at weak couplings, particles cannot entirely move freely along the $y$-direction, and they become highly localized in small regions. This degree of confinement is consistent with a $q2D$ scenario where the penetrable soft harmonic potential drives the order–disorder transitions (see snapshots of Figure 2b). This behavior is similar to that observed in [12–15], where the spreading of the peaks in the center of the channel indicates a loss of structure in that region.

By increasing the stiffness parameter to $k \approx 1.5$, the parabolic potential induces a transition from 4- to 3-strata configuration and evolves to 2-strata configuration when $k = 3$ (see snapshot in Figure 2c). When $k = 3$, the system exhibits two well-defined channels with no particles located at the center of the channel. However, when $k = 4$, the two-channel configuration is lost. In fact a second structural stage occurs in the interval $4 \leq k < 9$, see Figure 2c,d. For such $k$-values, there is a monotonic transition from a two-channel to a single-channel configuration, i.e., from $q1D$ to $1D$. The increase of the height of the peak and, simultaneously, the narrowing of the $P(y)$ distribution width signals the transition. These features are an indication that particles become more localized in the $y$-direction. However, in the interval $4 \leq k \leq 6$, we should carefully interpret the number of peaks as equal to the number of strata. For example, for $k = 5.0$, it seems that the system exhibits a three strata configuration, but in reality, this is not the case; it occurs that some regions of the box have single file particle configurations with particles located close to the box center, while other regions exhibit two-channel configurations. An analogous situation prevails for $k = 6$, see snapshot in Figure 1. As $k$ increases, the two-strata regions undergo a transition to a single-file configuration.

We now discuss the limit of strong couplings (Figure 2d). As $k$ increases, the density fluctuations relax faster in the $x$-direction, while in the $y$-direction, particles become highly localized. For $k \geq 9.0$, the system has practically reached a $1D$ configuration (see snapshot of Figure 2d), where the center of mass of the particles moves in a narrow region slightly larger than the particle radius. In Figure 2d, the probability distribution $P(y)$ also exhibits almost one single peak that allows us to conclude that the system behaves as in a $1D$ channel.

Following Reference [23], one can introduce a classification of the system dimensionality using the value of $L_y^{max}$ and the number of channels in the system. Then, a $1D$ system is such that $k \geq 9$, where $L_y < 2a$, i.e., a single-file configuration. A $q1D$ system is such that $2 < k < 9$, where the width along the $y$-direction is within the interval: $5a \leq L_y < 10a$. A $q2D$ system can be then defined when $0 < k < 2$ and the movement along the $y$-direction is restricted within the interval: $10a \leq L_y^{max} < 80a$. In general, the snapshots of Figures 1 and 2 provide an overview of the evolution of the dimensional transition as a function of the stiffness parameter. It is interesting to note that such transitions and the ordering (as we see further below) might be associated with the positional freedom that offers the external potential [6].

### 4.1.2. Structural Behavior

In the subsequent analysis, we include the case $k = 0$ (no external field) as the reference system to discuss the observed structural transitions. However, to characterize the orientational ordering in the interval $0.1 \leq k \leq 1$ ($q2D$ regime), we only consider those (blue) particles that belong to the bulk. Thus, in the calculations given by Equations (1) and (2),

we have excluded the (red) boundary particles. A similar approach was used in [12,14], where the colloids close the walls were treated differently than those in the bulk.

In the homogeneous system ($k = 0$), the long-range orientational order is absent, typical of a liquid phase. However, even for a small perturbation, such as $k = 0.1$, the system displays a dramatic increase in hexagonal order, similar to that predicted for a crystal. For this coupling, the system exhibits large space voids between the boundary particles and the available space in the $y$-direction. Hence, the creation of these voids results in more significant suppression of the local area fraction; particles get trapped in a reduced region compared with the original values of $L_x$ and $L_y$ leading to a highly packed system that exhibits hexagonal ordering. The correlation between hexagonal order and the presence of large voids adjacent to the wall has also been observed in circular optical corrals [16].

During the interval where the transition from $q2D$ to $q1D$ occurs, increasing the stiffness parameter not necessarily decrease the orientational order. In the interval, $0.5 \leq k \leq 1.0$, the $g_6$ exhibits a non-monotonous behavior. The orientational order decays for most of the $k$-values, except for $k = 0.8$ and $k = 1.0$; at these two particular values of $k$, the system displays long-range orientational order typical of a solid phase [37]. Note that in this situation, the boundary particles exhibit a high spatial order (see snapshot of Figure 3b). The ordering of red particles can be associated with promoting the hexagonal order of those particles that are not part of the boundary. However, for $k = 0.9$, this mechanism is interrupted since boundary particles are less ordered and cannot keep the system (bulk) in an ordered configuration (see snapshot of Figure 3b).

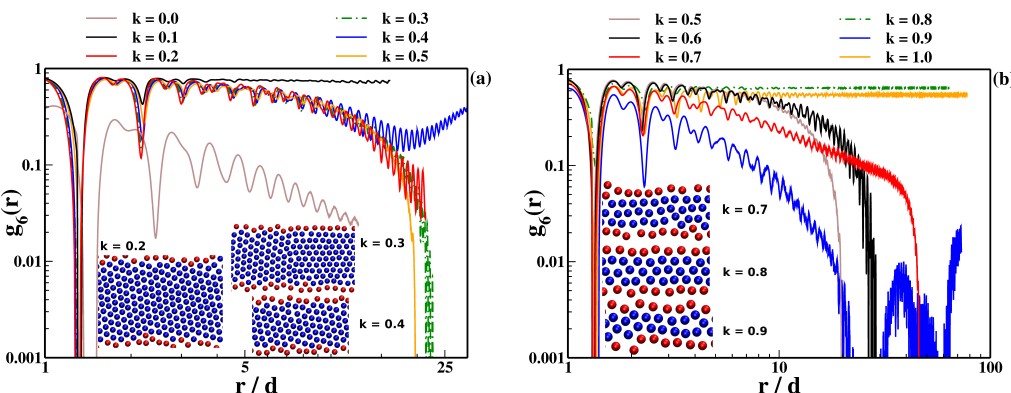

**Figure 3.** Orientational correlation function, $g_6(r)$, in the intervals (**a**) $0 \leq k \leq 0.5$ and (**b**) $0.5 \leq k \leq 1.0$, where one can easily distinguish between particles belonging to the boundary (red) and those confined to a certain region (blue).

One can notice an analogous order–disordered structural behavior in the pair distribution function, $g(x)$, where it seems that the system melts for $k = 0.9$, but is highly ordered for $k = 0.8, 1.0$ along the $x$-direction (see Figure 4b); this connection is discussed further below. Thus, varying the strength of the external potential, particles that belong to the bulk experience a dramatic change in their configuration, suggesting that the degree of confinement in combination with the softness of the harmonic potential induces melting and freezing transitions. This result is markedly different from that observed in $q2D$ superparamagnetic colloids where the hard wall imposed a different structure between the particles at the wall and those of the adjacent row. This inconmmensurability prevents the formation of a perfectly hexagonal structure [13].

A curved boundary only permits local hexagonal ordering when deformations have a low energetic cost. Thus, adaptive confinement promotes or inhibits hexagonal order by a variation of the stiffness parameter [16,21]. It seems that the ordered–disordered pattern found in the $g_6(r)$ is analogous to that observed in experiments with adaptive confinement using optical corrals. In our case, by controlling the harmonic potential softness, it is possible to induce transitions between ordered and disordered structures. For instance, a

hexagonal configuration can melt by a small increment in $k$. On the other hand, when the trap stiffness increases beyond $k = 1.0$, the system undergoes a transition from $q2D$ to $q1D$; thus, it is not longer possible to define an "inside region". In fact, for values $k > 2.0$, the system consists primarily of only two strips. Thus, it has no sense to calculate the $g6(r)$.

We also follow the evolution of the colloidal micro-structure through the one-dimensional pair distribution function and the static structure factor. We only present results in the direction perpendicular to the confinement, i.e., $x$-direction. For $k = 0$, when the external potential is absent, the condition $g(r) = g(x) = g(y)$ is fulfilled (the same applies for the static structure factor). Besides, both structure correlation functions exhibit the typical features of colloids interacting with long-ranged repulsive potentials: (i) particles do not feel the hard-core interaction, (ii) the system shows a typical fluid-like order, and (iii) the structure factor main peak has a height below 5.5, and there is a characteristic length scale given by the average distance $d = \rho^{-1/2}$. However, when the external potential is turned on, one observes variations in the local structure, even for small values of $k$, see Figure 4a.

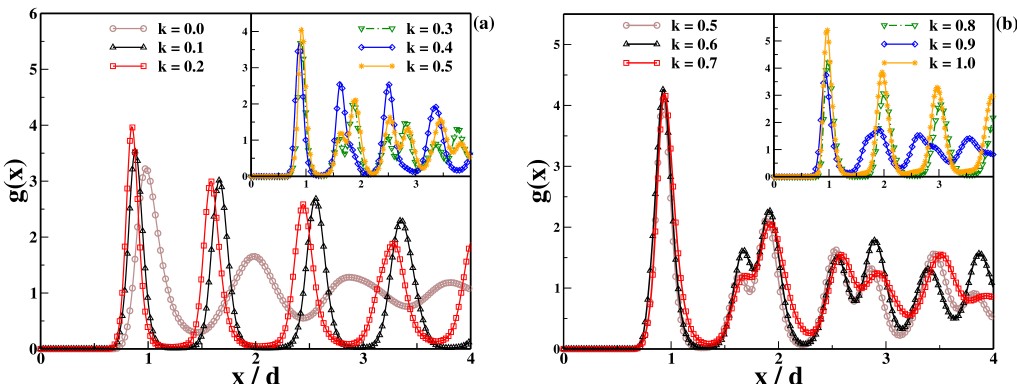

**Figure 4.** Pair distribution function along the $x$-direction for several values of the stiffness parameter: (**a**) $0 \leq k \leq 0.5$ and (**b**) $0.5 \leq k \leq 1.0$.

The $g(x)$ quantifies the order along the direction without confinement and makes evident the particle spatial reordering when $k$ is varied. For example, for $k = 0$, particles are preferentially located at multiples of the distance $d$. However, when $k > 0$, particles try to form a hexagonal lattice separated by a distance $\approx 0.90d$, which is close to the exact value of $\sqrt{3}/2$ corresponding to the separation between two rows of particles in a perfect hexagonal lattice. The static structure factor $S(q_x)$ corroborates this observation (see Figure 5a). For $k = 0$, the characteristic length scale is determined by $d$; hence, the location of the main peak of $S(q_x)$ is at the position $q_x \approx 2\pi$. In contrast, for $k = 0.1$, the location peak is moved to the right, showing that particles are closer, forming more compact structures as compared with $k = 0$.

For $k = 0.2, 0.4$ particles are located at positions that resemble the hexagonal order. In fact, along the unconfined direction, they are closer than those in the case $k = 0.1$. The $S(q_x)$ (Figure 5a) allows one to notice this feature, where the width of the main peak becomes highly narrow, and the other peaks locate at multiples of the characteristic length; the latter can be obtained from the expression: $ql_c = 2\pi$. Thus, the characteristic length is $l_c \approx 0.83d$. However, local and long-range order in the $y$-direction is lost (data not shown). These features could explain the loss of orientational correlation at very long distances (Figure 3a). Note that the structure factor is also sensible to the order–disorder state of the boundary particles; see snapshots of Figure 3a.

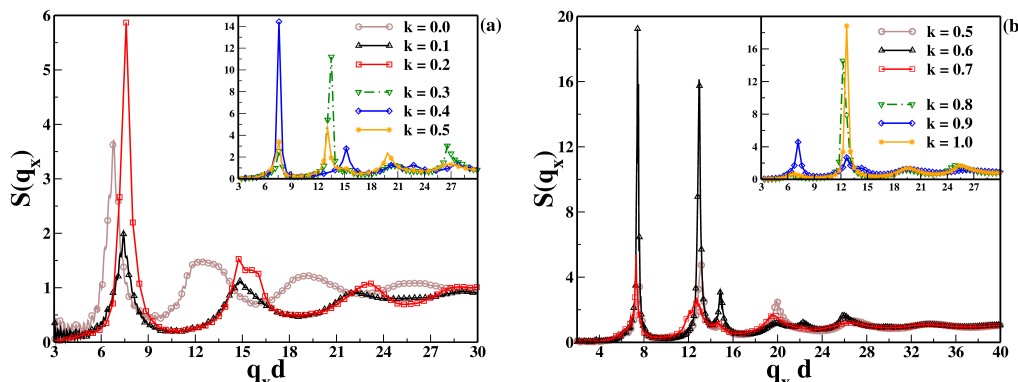

**Figure 5.** Static structure factor along the *x*-direction for several values of the stiffness parameter: (**a**) $0 \le k \le 0.5$ and (**b**) $0.5 \le k \le 1.0$.

For $k = 0.3$ and $0.5, 0.7, 0.9$ the $g(x)$ exhibits a loss of correlations at large inter-particle separations (Figure 4a,b). The two first nearest neighbors locate at multiples of the average distance $d$, but for larger distances, the distribution of peaks does not exhibit a clear periodicity, and the height of the peaks has decreased. The structure factor corroborates this feature, where the first peak has decreased its height, indicating that another characteristic length is emerging (see Figure 5a,b). From these features, it seems that the system melts in the *x*-direction. Interestingly, the $g_6(r)$ captures this loss of correlation along the unconfined direction as a loss of hexagonal order at long distances. This scenario is consistent with the kind of softness-induced melting discussed above and illustrated by the disordered state of the boundary particles; see snapshots for $k = 0.7, 0.9$ in Figure 3b.

In clear contrast, for $k = 0.8$ and $k = 1.0$ the $g(x)$ and the $S(q_x)$ exhibit characteristics of a solid-like ordering. The $g(x)$ peaks locate at well-defined positions, multiples of the average distance, and the correlations become long-ranged. The $S(q_x)$ displays practically one highly narrow main peak, a signal of a dominant length scale in the colloidal suspension. This scenario is also consistent with the softness-induced freezing; see snapshots of Figure 3b. Besides, one should note that the main peak of the structure factor for $k = 0.2, 0.4, 0.6, 0.7, 0.9$ is located at different positions as compared with the main peak for $k = 0.8, 1.0$. This evidence that different natural length scales are present and the underlying structural ordering is different in each case. The peaks of the $S(q_x)$ for $k = 0.6$ deserve additional comments. For this case, the $g(x)$ exhibits a clear split at relatively short distances (Figure 4b). These peaks are associated with the second and third peaks of the $S(q_x)$, whereas the first peak is linked to the strong oscillations that the $g(x)$ displays at long distances (data not shown). These results are in agreement with those reported for superparamagnetic colloid in hard walls [12,14], where the presence of well-defined strata is associated with a more considerable degree of order. The authors also reported a re-entrant behavior of the system, transitioning from liquid-like to solid-like and back as the channel width varies.

For $k = 1$ and $3$ ($q1D$ regime) the colloidal system exhibits well defined 4- and 2-strata configurations, respectively (Figure 2c). For these values of the stiffness parameter, the $g(x)$ exhibits characteristics of a highly ordered system (Figure 6a). The peaks become sharper, reflecting that particles are more localized, and they are spatially correlated at long-distances, i.e., there are strong oscillations at several mean inter-particle distances, $d$. Moreover, there is only one characteristic length scale in the system, determined by the inter-maxima separations, i.e., the maxima are at positions that are multiples of $d$, and the valleys separate them, i.e., regions in the pair distribution where $g(x) = 0$. These valleys suggest that particles are highly localized and strongly correlated along the channel [25]. The static structure factor also reflects this ordering, where a highly narrow peak is at $qd \approx 2(2\pi)$ (Figure 6c).

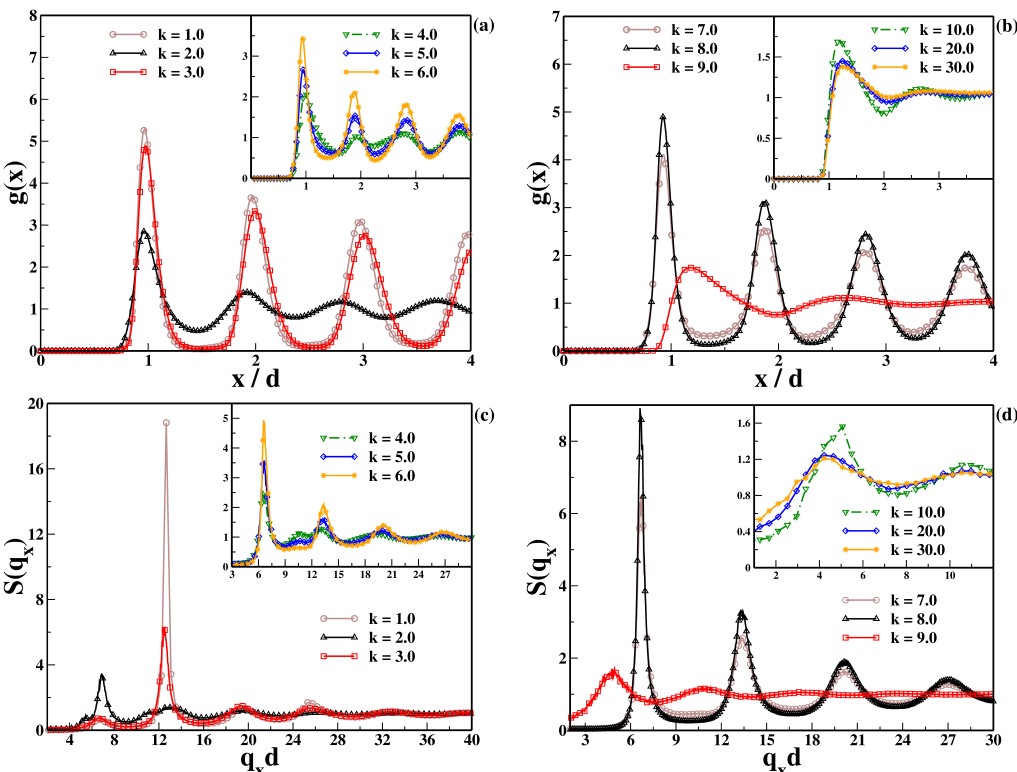

**Figure 6.** (**a**,**b**) Pair distribution functions and (**c**,**d**) static structure factors along the *x*- direction for several values of the stiffness parameter, $1 \le k \le 30$.

Interestingly, at the intermediate value of $k = 2.0$, the system exhibits a completely different structural scenario, see Figure 6a,c. The first peak of the $g(x)$ is located at shorter separations and there is a loss of correlation at longer distances. We link this structural reentrance with the transition from 3- to 2-strata. During this transition, there are no well-defined channels; in some regions, the system consists of two channels, and in other regions, there are clusters of particles forming three non-permanent channels. This structural scenario is the reason why one observes the third peak in the $P(y)$ (see Figure 2c). For this case, the short-range correlations observed in the $g(x)$ are seen directly in the $S(q_x)$, which shows smooth features, and the maxima positions represent the interplay among different length scales. The dimensional transition correlates with a structural change. The colloidal dispersion goes from a cooled-liquid state to a liquid state linked with the loss of correlations along the *x*-direction.

In the confinement regime corresponding to *k*-values in the interval $4 \le k \le 8$ (Figure 6), the system undergoes a monotonic increase in the correlations giving rise to another re-entrant freezing-like transition. One can notice that as *k* increases, the height of the peaks in the $g(x)$ and $S(q_x)$ also increases, and long-range correlations start to appear. This behavior is fully consistent with the one observed for repulsive $1D$ colloidal systems as a function of the potential strength, or packing fraction [25,28,38]. Hence, by increasing *k* from 4.0 to 8.0, particles become more confined along the *y*-direction, but also in the *x*-direction. Thus, although particles locate at well-defined positions, the periodicity does not correspond to the mean inter-particle distance, *d*. We associate this behavior with the structural transition in which the system goes from 2- to 1-channel configuration (see snapshots of Figure 2).

For values of the stiffness parameter $k \ge 9$, the system forms a single chain of particles. At this transition, the $g(x)$ acquires a much less pronounced structure (Figure 6b). In particular, the first peak dramatically decreases its height, there is a complete loss of correlations at long separations, and there is not periodicity along the channel. These features are typically associated with the structure of colloidal dispersions in a fluid phase.

The snapshot when $k = 9$ in Figure 1 visually corroborates that the limit of a $1D$ diluted system has been finally reached [25]. The inset of Figure 6d shows the structure factors for $k > 10$. Such $S(q_x)$ corresponds, accordingly to the $P(y)$ shown in Figure 2d, to $1D$ systems. We also corroborate that the structural behavior for $k \geq 10$ is very similar to that of a $1D$ colloidal system at the same density.

The physical implication from the above analysis is that varying the boundary stiffness alters the static properties of a confined material. Analogous to the observed behavior of a $3D$ hard-sphere dispersion under the action of a parabolic trap, we associate the re-entrant and phase-separating behaviors to the soft confinement. The penetrable harmonic potential offers more positional degrees of freedom than the typical rigid confinement, making an important energy contribution to the system free energy, which depends on the particle configuration [6]. Similar oscillating behavior in the global structure as a function of the degree of confinement has already been observed in $q2D$ superparamagnetic colloids between hard walls [12–15]. The oscillations in the structural properties indicate that the structure of the crystal can be altered by slight changes in the channel width.

### 4.2. Dynamics

To understand the particle transport during the structural transitions described in the previous section, we focus on dynamical correlations, such as the mean-square displacement and the self-part of the intermediate scattering function, for different values of the trap stiffness $k$. In addition to the non-monotonic structural behavior, transport phenomena should also exhibit peculiarities linked to the structural variations analyzed above. Open systems rarely exhibit such dynamical features.

Figure 7a shows the MSDs in the interval $0 \leq k \leq 0.5$. For the homogeneous system ($k = 0$), particles exhibit normal diffusion; the MSDs along the $x$- and $y$-directions, $W_x$ and $W_y$, respectively, are almost identical (within the statistical uncertainties), and both exhibit a linear time dependence at short and long times. When $k > 0$, the particle diffusion decreases in both directions, however, one can observe a faster diffusion when $k = 0.3$. The non-monotonic diffusive behavior observed during the interval from $k = 0$ to $k = 0.5$ is attributed to the different structural scenarios, where the cage formed by its nearest neighbors affects particle movement [39]. Inset of Figure 7a shows the MSD along the confined direction. The external field induces particle localization in such a direction, promoting a diffusion decrease of one order of magnitude. However, at long times, the dynamical behavior is practically the same regardless the stiffness parameter value. It is clear that the variation of the confinement conditions affects the diffusion mechanisms and their magnitude in both directions. In particular, it can induce either faster or slower particle diffusion.

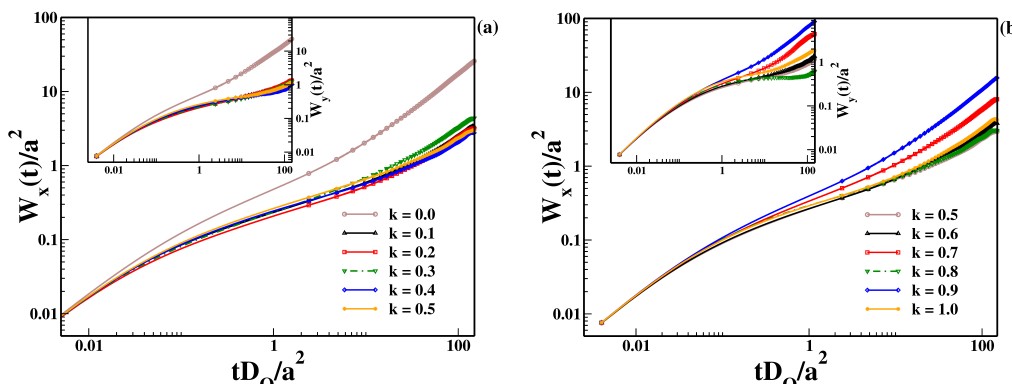

**Figure 7.** $W_x(t)$ for different values of the stiffness parameter, $k$, to cover low couplings. (**a**) $0 \leq k \leq 0.5$, (**b**) $0.5 \leq k \leq 1.0$. Inset shows the MSD in the $y$-direction.

The MSDs at long times for $k = 0.5, 0.6, 0.7$, increase monotonically with $k$ (Figure 7b). Such increase might be related to the new lengths scales that emerge in the system induced

by the boundary particles at the order–disorder transitions. However, for $k = 0.8, 1.0$, there is a substantial decrease in particle diffusion. From Figure 3b and the inset of Figure 5b, we recall that both scenarios exhibit the same structural behavior, which results in a similar slow diffusion. From the inset of Figure 7b, one can observe that for $k = 0.9$ (blue line), the flexible confinement allows larger displacements in the $y$-direction resulting in a faster diffusion along the $x$-direction. Hence, the origin of the slow–fast–slow diffusion mechanisms might also be related to the boundary particle rearrangements, leading to different solid- and liquid-like structures (see snapshots of Figure 3b). In References [12,14] a similar dynamical behavior was reported, where the defect concentration (disorder) is intimately tied to the dynamics of the system; the transport properties of the colloidal particles also oscillate as a function of the dimensionless channel width. The MSDs in the confinement direction also exhibit non-monotonic behavior. It mimics the dynamical scenario along $x$-direction. Note that despite the structure and dynamics for $k = 0.8$ and $k = 1.0$ along the $x$-direction are very similar, the $W_y(t)$ for $k = 1$ is evidently larger than $k = 0.8$. This behavior could be interpreted as evidence of directional dependent melting [10], where the system melts in one direction but not in the other.

For intermediate couplings, particularly, $1 \le k \le 5$, see Figure 8a, the long-time behavior of the MSDs in the $x$-direction displays a non-monotonic behavior with $k$. For example, for $k = 1$ particles diffuse very slow in contrast with the other values of $k$, except for $k = 3$, which exhibits a similar structural and dynamical $k$-dependence. The absence of a monotonic dynamical behavior is related to the structural transition that the system undergoes, i.e., from 4-strata ($k = 1.0$) to 2-strata ($k = 3.0$). The transition from $k = 1.0$ to $k = 2.0$ is signaled by a re-entrant structural behavior associated with disordered particle configurations. This scenario allows the particles to move more freely along the $y$-direction. The loss of translational correlations reflects in the increase of the diffusive behavior in both directions.

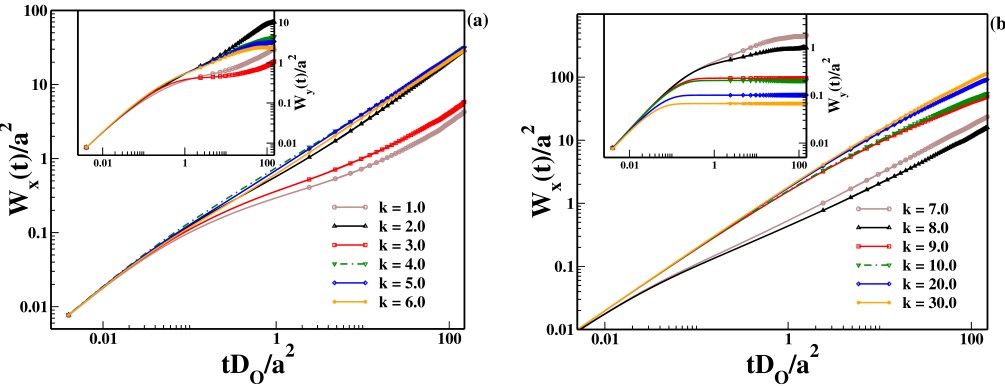

**Figure 8.** $W_x(t)$ for different values of the stiffness parameter, $k$, to cover intermediate and strong couplings. (**a**) $1 \le k \le 6$; (**b**) $7 \le k \le 30$. Inset displays the MSDs in the $y$-direction.

Particular attention deserves the case $k = 3.0$; there, the system has reached a configuration of two channels (see snapshot of Figure 2c), where boundary particles cannot be defined. Compared with $k = 2.0$, the system exhibits higher structural order along the $x$-direction and a completely different characteristic length scale (Figure 6a,c). The two-strata configuration creates an energetic barrier, which complicates the particle exchange. These structural features are then responsible for the slower diffusion of particles in both directions (red line of Figure 8a).

In the interval, $4 \le k \le 8$, it is possible to observe a monotonic decrease in the diffusive behavior (Figure 8) in both $x$ and $y$ directions. During this interval of $k$-values, the rearrangement of particles is always in zig–zag configurations. The particle displacements in the $y$-direction, concerning the simulation box center, restrict the motion of neighboring particles along the $x$-direction. Thus, as $k$ increases, particles become more localized in both directions resulting in a systematic slowing down of the diffusion in any direction.

Approximately, for stiffness parameter values of $k \geq 9$, the long-time behavior of the $W(x)$ exhibits a monotonic increase. To understand this, we should think in terms of a $1D$ single-file system. When $k$ increases, $L_x$ and $L_y$ increases and decreases, respectively, resulting in slightly higher particle mobility in the $x$-direction (Figure 8) and suppression of particle motion in the $y$-direction (see inset of Figure 8b). We expect that values of $k > 30$ will suppress particle fluctuations along the $y$-direction, and the system will reach the limit of a true $1D$ system. In this limit, the dynamics at very long times should be characterized by the relation $W_x(t) \propto t^{1/2}$ [25,28,38].

For the highest value of $k = 30$ explored in this work, the correlation functions $g(x)$ and $S(q_x)$, see Figure 6b,d, respectively, are practically identical to the true $1D$ system (data not shown). However, the long-time dynamics do not follow the single-file diffusion behavior [25,28]. Given that, strictly speaking, a $1D$ system with a packing fraction of $\sim 0.17$ is a highly diluted system [25], the time needed to reach such a sub-diffusive regime is larger than the time window explored in the simulations. In other words, particles must displace longer distances to develop the translational correlations that give rise to the single-file dynamics behavior. Hence, we claim that the observed behavior in Figure 8b is only part of the transitional regime to the sub-diffusive non-Fickian behavior.

Particle dynamics are also characterized through the self-part of the intermediate scattering function. The delicate interplay between particle–particle and particle–potential interactions results in rich structural scenarios, which leads to multiple temporal relaxation regimes. Of course, we associate this behavior with the soft nature of the confining potential. The sISF decays exponentially to zero in an ergodic system; when $k = 0$, the sISF exhibits this behavior in both directions (circles in Figure 9a). In contrast, when the external field acts on the particles, even for weak couplings (Figure 9a,b), the sISF in the non-confinement direction barely reaches zero within the time window of the simulations. The sISF decays in a non-monotonic way, which corroborates the observed structural and dynamical features in the $g(x)$, $S(q_x)$, and the MSD.

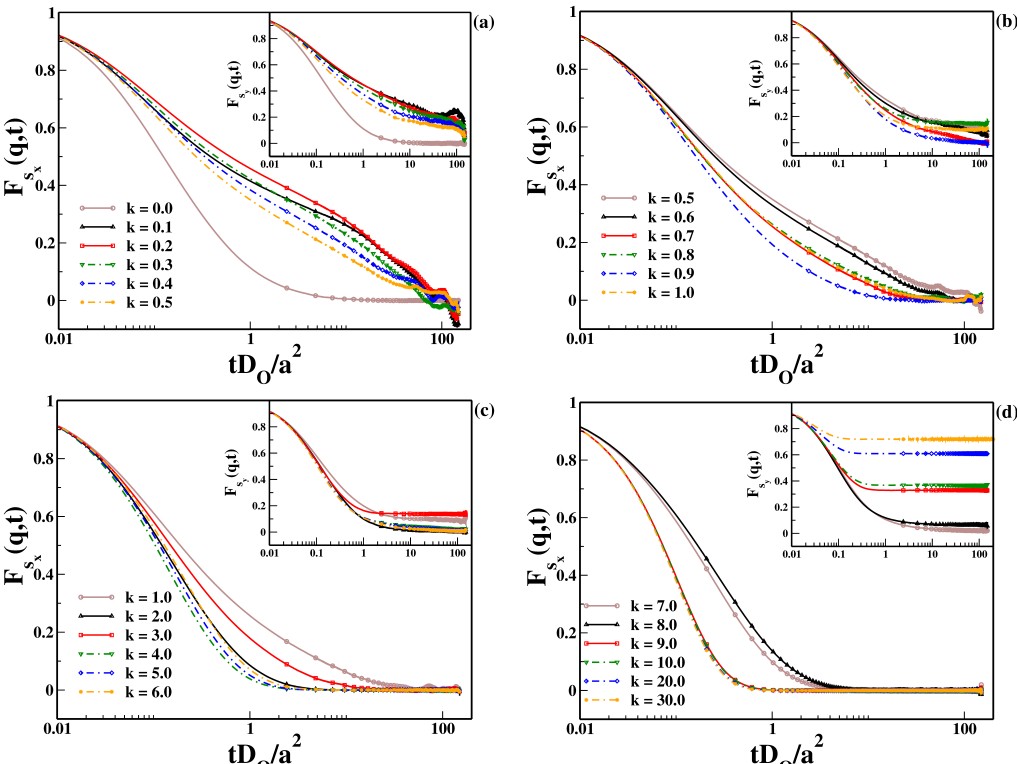

**Figure 9.** Self-intermediate scattering functions $F_s(q = \pi, t)$ as a function of the trap stiffness, going from (**a**) and (**b**) weak to (**c**) intermediate and (**d**) strong couplings. The main figure shows the sIFS along the non-confinement direction, and the inset displays the sIFS in the $y$-direction.

From the inset of Figure 9a, one can observe that the sIFS in both directions exhibit the same trend. In contrast, for $k = 0.8$ and $k = 1.0$ the sIFS$_y$ (Figure 9b) exhibits a plateau. This plateau is a signature of high localization and suppression of particle transport along the confinement direction. From Figure 9a,b is clear that local structure is influenced even at weak coupling with significant implications for the dynamics along the $x$ and $y$ directions.

In the intermediate and strong coupling limit, $k > 1$, one can observe a clear distinction in the colloidal system ergodicity. The sIFS$_x$ decays to zero for all cases (Figure 9c,d). In clear contrast, the sIFS$_y$ exhibits a plateau at shorter times, indicating that particles become more confined as the coupling with the external potential is increased. Besides, for systems with $k \geq 9.0$, the sIFS$_x$ decays much faster than systems with lower values of $k$. This faster relaxation is associated with the transition from $q1D$ to a diluted $1D$ system where particles exhibit higher mobility (Figure 8b).

In general, structural ordered states are associated with lower diffusion. One can then establish a connection between the slow or fast relaxation of particles with features such as long-range particle correlations, long-range orientational order, dimension variations, and slow and fast diffusion. This connection is better summarized in the $k$-dependence of the structural relaxation time $\tau_\alpha$ [16,26,40]; $F_s(q, \tau_\alpha) = e^{-1}$, along the $x$-direction (Figure 10a). As can be seen from the figure, the $\tau_\alpha$ displays multiple relaxation regimes, which depend strongly on the degree of confinement.

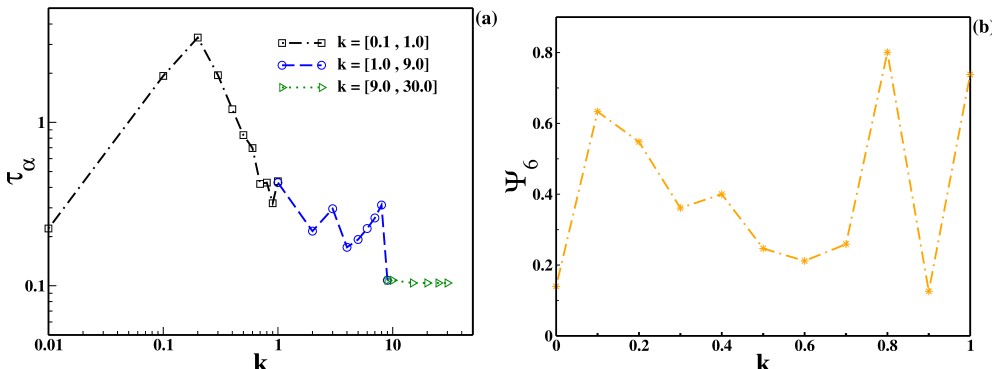

**Figure 10.** (**a**) Structural relaxation time and (**b**) global orientational order parameter as a function of the stiffness parameter, $k$.

We split the $k$-dependence of $\tau_\alpha$ into three regimes to highlight the different stages of particle relaxation. The first stage, i.e., weak couplings, allows us to see the important role of the larger positional freedom of boundary particles on the particle relaxation. There, the order–disorder inter-converting configurations induce an extended relaxation process associated with high ordered states. Systems such as $k = 0.1, 0.8$ and $1.0$ that exhibit long-range orientational order characterized by the high value of the orientational global parameter $\Psi_6$ (see Figure 10b) also exhibit larger relaxation times. Thus, it is possible to induce freezing and melting scenarios in confined systems by altering the boundary properties.

On the other hand, in the regime $1.0 < k < 4.0$, there are slow and fast modes of relaxation associated with the dimensional transition from $q2D$ to $q1D$. The peak at $k = 3$ is due to the transition from 3- to 2-strata; here, the colloidal system displays two-well-defined channels with high structural order. In the regime $4.0 \leq k < 8.0$, the systematic increase in $\tau_\alpha$ is related to the slow dynamics behavior during the dimension transition from $q1D$ to $1D$. The peak at $k = 8.0$ is associated with the fact that the colloidal system almost reaches the $1D$ file configuration. The box length in the $y$-direction prevents the mutual passage of particles inducing a highly ordered configuration along the $x$-direction; thus, the structure factor exhibits characteristics typical of a solid-like state [25]. Finally, in the last regime, $k \geq 9.0$, there is only one relaxation time, which does not depend on the stiffness parameter anymore. This change in the temporal relaxation behavior occurs at the onset of a dimensionality change. At this stage, the system has reached the single-file configuration.

Thus, relaxation occurs faster, which means that at the same particle concentration, the particle transport process occurs faster in $1D$ than in $2D$.

## 5. Final Remarks and Perspectives

We have theoretically studied the physical properties of charged colloidal dispersions under soft confinement. In the model system, each particle experiences a Hookean restoring force. At a given instant, particles at the boundary acted as a flexible wall capable of affecting the interior (bulk particles) adaptively. Hence, as the boundary particles experienced order–disorder configurations, a wide variety of structural and dynamical scenarios could be accessed that are not present in conventional static confinement.

A strong link between the particle dynamics and the structural changes induced by the external potential was highlighted. By systematically varying the coupling with the external field, it was possible to manipulate the self-assembly and the transport processes of the interacting colloids. Even in the weak-coupling regime, the harmonic potential dramatically affected the rearrangement of the colloidal particles. Given that particles at the boundaries did not interact with image particles or a rigid wall, they seemed to be especially sensitive to the variation of the stiffness parameter $k$. These particles played a crucial role in inducing order or disorder to those particles in the bulk.

The probability distribution, $P(y)$, allowed us to describe and categorize the transition from the homogenous $2D$ colloidal system until it reaches the final $1D$ configuration. For values up to $k = 1$, we characterized the orientational ordering with the bond-angular correlation function $g_6$. We found that the presence of the harmonic potential promotes hexagonal ordering that otherwise will be absent. However, such an orientational ordering exhibited a non-monotonic dependence on $k$. There was a delicated interplay between the particle–particle repulsions and the confining potential. For $k = 0.8$ and $1.0$, the orientational ordering approached a constant value for large distances; this behavior resemblances the existence of a solid or crystalline phase.

In the interval $0 < k < 9$, observables such as the pair distribution function and the static structure factor successfully captured the numerous ordered and disordered transitions that occurred as a function of $k$. These observables not only revealed the re-entrant structural behavior or the solid- or liquid-like state, they also allowed us to identify differences in the underlying structural order.

The dynamical behavior strongly correlated with the structural scenarios that emerged as $k$ was gradually increased. The colloidal system exhibited lower diffusion for those values of $k$, where long-range orientational order or solid-like behavior was observed. In contrast, the MSD in both $x$- and $y$-directions exhibited enhancement diffusion where a re-entrant liquid-like transition seemed to occur. The MSD along both directions nicely captured the transition to the $1D$ regime. When the system reached the single-file configuration, the diffusion along the confined direction was completely suppressed while it increases along the unconfined direction.

The self-part of the intermediate scattering function corroborated the structural and dynamical features, namely, the sIFS exhibited a non-monotonic decay associated with the freezing and melting scenarios present in the interval $0 < k \leq 3$. The sIFS also captured the $1D$ regime where it exhibited a plateau in the confined direction and faster decay along the $x$-direction. The dynamical behavior complexity was summarized through the structural relaxation time, which displayed multiple relaxation times accordingly with the structural scenario and the degree of confinement.

As discussed above, we paid particular attention to weak and intermediate particle–trap couplings, where the structural arrangement of boundary particles was able to induce, for example, melting- or freezing-like states and faster or slower particle diffusion. The effect on the static and dynamic properties of colloidal dispersions of these peripheral particles as a kind of adaptive boundary deserves to be studied both theoretically and experimentally in more detail. Work along this line is in progress.

**Author Contributions:** All authors contributed equally to this work. R.C.-P. conceived the study. E.C.E.-D., and S.H.-V. implemented the simulation model, performed the data adquisition and prepared the manuscript. All authors have read and agreed to the published version of the manuscript.

**Funding:** This work was partially supported by the "Odysseus" Program of the Flemish Government, the Flemish Science Foundation (FWO-Vl), PRODEP, CONACyT (Grant Nos. 237425 and 287067), NPTC-PROMEP 6613/2013, and Universidad de Guanajuato. R.C.-P. also acknowledges financial support provided by the Marcos Moshinsky Foundation.

**Institutional Review Board Statement:** Not applicable.

**Informed Consent Statement:** Not applicable.

**Data Availability Statement:** Data is contained within the article.

**Acknowledgments:** Authors also thank the Xalapa Superior Institute Technology for providing computational resources and the General Coordination of Information and Communications Technologies (CGSTIC) at Cinvestav for providing HPC resources on the Hybrid Cluster Super-computer "Xiuhcoatl", which have contributed partially to the research results reported in this paper.

**Conflicts of Interest:** The authors declare no conflict of interest. The funders had no role in the design of the study; in the collection, analyzes, or interpretation of data; in the writing of the manuscript, or in the decision to publish the results.

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
