# Peer review of "Ordering and Dynamics of Interacting Colloidal Particles under Soft Confinement"

_colloids, doi:10.3390/colloids5020029_

Round 1

Reviewer 1 Report

In this article, Herrera-Velarde, et al. studied the structural transition of a two-dimensional colloidal system subjected to a one-dimensional parabolic potential trap. The results are well presented and analyzed in great depth following standard colloidal physics literature. However, there are critical concerns about the methodology that the authors need to clarify and demonstrate carefully before the results can be published. Comments of this manuscript are listed below:

Major comment:

  1. About the change of the box dimension during the simulation. On Line 112, the authors mentioned that “the maximum displacement of particles in the y-direction” is selected as the updated box length Ly. Could the authors elaborate on why changing the box size is necessary? The structural transition and ordering could happen simply corresponding to the dimension change in the simulation box and the author need to be clear about which parts of the conclusion can be attributed to the softness of the potential. The simulation should also be carried with a fixed dimension of the simulation box, as the transition can also happen when the simulation box size changes from 2D to quasi-1D. The authors need to check this critical control to support their arguments

Minor comments:

  1. For figure 1, it seems the symbol k means k* in the figure, but it means non-normalized k in the x-label. It is suggested to keep all symbols consistent and simply use k* when it is normalized.
  2. The symbol of stiffness parameter “k” is commonly used to denote wave vector. It’s better to add a subscript to avoid confusion.

Reviewer 2 Report

Salvador et al, brought this interesting research paper to study the physical properties of charged colloidal dispersions under soft confinement. In the model system, each particle experiences a Hookean restoring force. The paper is well written and overall in a good quality. I am happy to recommend to accept to publish after addressing the following point:

  1. The definition of 'soft confinements' is not very clear, author should provide some illustrations to describe them. and also described the origin and refer them to the experimental parameters/phenomenon.

Round 2

Reviewer 1 Report

Thanks for the careful and thorough discussion the authors made to rationalize their observations and simulation design. Setting appropriate boundary conditions for studying soft potentials is a complicated question and I believe the approach the authors developed has its unique value. The authors have adequately addressed my concerns and I recommend the publication of this article.